# Fatigue Performance of High- and Low-Strength Repaired Welded Steel Joints

**Jan Schubnell [1,\*], Phillip Ladendorf [2], Ardeshir Sarmast [1], Majid Farajian [1] and Peter Knödel [2]**

[1] Fraunhofer Institute for Mechanics of Materials (IWM), 79108 Freiburg, Germany; ardeshir.sarmast@iwm.fraunhofer.de (A.S.); Farajian@slv-duisburg.de (M.F.)

[2] Steel and Leightweight Structures, Karlsruhe Institute of Technology, 76131 Karlsruhe, Germany; philipp.ladendorf@kit.edu (P.L.); peter.knoedel@kit.edu (P.K.)

[\*] Correspondence: jan.schubnell@iwm.fraunhofer.de; Tel.: +49-761-5142-235

**Abstract:** Large portions of infrastructure buildings, for example highway- and railway bridges, are steel constructions and reach the end of their service life, as a reason of an increase of traffic volume. As lifetime extension of a commonly used weld detail (transverse stiffener) of these structures, a validated approach for the weld repair was proposed in this study. For this, welded joints made of S355J2+N and S960QL steels were subjected to cyclic loading until a pre-determined crack depth was reached. The cracks were detected by non-destructive testing methods and repaired by removal of the material around the crack and re-welding with the gas metal arc welding (GMAW). Then, the specimens were subjected to cyclic loading again. The hardness, the weld geometry, and the residual stress state was investigated for both the original- and the repaired conditions. It was determined that nearly all repaired specimens reached at least the fatigue life of the original specimen.

**Keywords:** fatigue; repair welding; residual stress; non-destructive testing





## 1. Introduction

A large portion of infrastructural buildings, such as bridges and offshore structures, are steel constructions. Especially railway or highway bridges are exposed to a strongly increasing rail cargo and traffic volume. This leads to an increasing load of such structures and thus, to an increasing fatigue damage. In many cases, weld details are especially affected as a reason of their comparably low fatigue resistance. In the Federal Republic of Germany, nearly 50% of the steel bridges are built before 1980 [1] and around 55% of the railway bridges are built before 1950 [2]. Similar conditions are reported from the United State of America where around 85% of the bridges in Minnesota were built before 1986 [3]. Originally, these structures are designed for significant lower loads and a higher number of repair cases is expected for the future.

The right choice of the repair strategy is an important factor to extend the fatigue life of cracked steel structures [3]. Usually, fatigue cracks are detected by visual inspection or non-destructive testing methods in periodically time intervals. After the detection, possible repair methods are the usage of bolted splices or by gouging and re-welding the material [4]. However, bolted splices are not always efficient for relatively minor fatigue damage especially if the available working space around the damage is limited [5]. Compared to the conventional bolt splice method, repair welding is a more cost-efficient solution and can be performed with less time effort. However, possible weld-defects like splatters, undercuts or cold laps can be induced by re-welding and might only lead to a comparatively small, extended life-span after the retrofit.

Due to countless different weld-details present in steel-bridges and -constructions, many repair cases exist [4,6–12]. Especially, the IIW-document XIII-2284r1-09 [4] contains a large number of repair-cases from the 1960s. Based on many of these cases, it has to be pointed out, that a retrofit of fatigue-damaged steel structures is often carried out by

combining many of the before mentioned measures. Hence, it is not possible to quantify the effect of every single one of them. In order to quantify the sole effect of repair-welding, a literature survey has been carried out. Wylde [13] carried out fatigue experiments on transverse and longitudinal stiffeners with fillet welds in the as-welded state until a pre-defined crack length and depth had been reached. A subsequent repair was prepared by removing the damaged material by disc grinding. The repair-welding was fabricated by electrode welding. He concluded that the original fatigue detail class can be restored in case there are no internal defects present.

The fatigue behavior of repair-welded transverse stiffeners were also investigated in [14–16]. However, the repair welding was performed at unloaded and uncracked specimen. Thus, no direct comparison of the welded joints in original condition and in repaired condition was possible.

In the scope of the German FOSTA research project P864 [17], fatigue experiments were conducted at butt welds and specimens with longitudinal stiffeners and fillet welds. In the case of the butt welds, the repaired weld toes could not be tested up to a fatigue failure due to a failure of a non-repaired weld toe. Therefore, no conclusion for the butt welds can be drawn. Instead of this, the fatigue strength of the repaired longitudinal stiffeners has been increased compared to the as-welded condition, since the weld angle was smaller after the repair.

Different, but few studies have shown that the fatigue life of welded joints could be significantly extended by gouging and re-welding, subsequently named as repair welding [4,13,18–20]. However, these experimental studies were not included in current design codes from the International Institute of Welding [21], Eurocode 3 [22] or the German FKM guideline [23]. For this reason, a validated approach for the repair of welded structures is needed to assure that a maximum possible fatigue life by re-welding could be reached. Furthermore, the fatigue classes (FAT) of such repaired weld details needs to be reported so they can be included for the design and fatigue life estimation of repaired welded steel constructions.

The aim of this work was to develop a validated approach for repair welding of fillet welds. Double sided transverse stiffeners were chosen as corresponding weld detail. For this, a stepwise procedure containing the fatigue test in original condition, crack detecting by NDT methods, gouging, multi pass re-welding and fatigue test in repaired condition were performed.

## 2. Materials and Specimens Detail

Two commonly used steel grades were investigated with significant different mechanical properties. As mild, structural steel S355J2+N in normalized condition with a yield of 402 MPa was used. Additionally, the high strength, quenched and tempered steel S960QL with a yield of 1011 MPa was used for this work. Fatigue tests were performed on 88 specimens. The chemical compositions of the base materials are shown in Table 1 and the mechanical properties are given in Table 2. The chemical composition was measured by spectral analysis.

**Table 1.** Chemical composition of the investigated base materials.

| Materials | Elements (wt.%) (Fe = bal.) | | | | | | | | | | | | | |
|---|---|---|---|---|---|---|---|---|---|---|---|---|---|---|
| Element | C | Mn | Si | P | S | Cr | Ni | Mo | V | W | Cu | Al | Ti | CEV * |
| S355J2+N | 0.161 | 1.47 | 0.17 | 0.0107 | 0.0053 | 0.040 | 0.035 | 0.007 | 0.008 | 0.004 | 0.015 | 0.032 | 0.0125 | 0.42 |
| S960QL | 0.155 | 1.23 | 0.20 | 0.0095 | 0.0017 | 0.194 | 0.084 | 0.599 | 0.046 | 0.007 | 0.013 | 0.057 | 0.003 | 0.53 |
| G4Si1 ** | 0.08 | 1.65 | 1.0 | - | - | - | - | - | - | - | - | - | - | - |
| Mn2NiCrMo ** | 0.10 | 1.80 | 0.80 | - | - | 0.350 | 2.300 | 0.600 | - | - | - | - | - | - |

* According to DIN EN 10025-2, ** data sheet.

**Table 2.** Mechanical properties of the investigated base materials.

| Materials | Yield Strength (MPa) | Ultimate Strength (MPa) | Elongation (%) | Hardness (HV10) | Generic Name |
|---|---|---|---|---|---|
| S355J2+N | 402 | 538 | 25 * | 169 | - |
| S960QL | 1011 | 1060 | 14 * | 316 | Strenx S960E |
| G4Si1 * | 390–490 | 510–610 | ≥25 | - | SG3 |
| Mn2NiCrMo * | 880–920 | 940–980 | 16–20 | - | Union X90 |

* data sheet.

In this work, the investigated fillet welds were manufactured with the gas metal arc welding (GMAW) process. The weld detail was a double-sided transverse stiffener. The specimens drawing is shown in Figure 1 and the cross section of the actual welds are shown in Figure 2. In total, three sections per material were extracted around the center of the welded sheets. Samples were hot mounted at 180 °C under 30 kN using an automatic metallurgical mounting press Struers LaboPress-3. Heating and cooling times lasted respectively 8 and 5 min. The polishing is realized on a Struers Tegramin-30 polishing machine using the following felts: Metall MD-Allegro 9 μm, Metall MD-DAC 3 μm, Metall MD-Chem 0.1 μm for further investigations. Some higher differences of the welds shape are visible for S960QL compared to S355J2+N but this could not be observed for every extracted section. The length of the welded joint was 1250 mm for S355J2+N and 1500 mm for S960QL. The manufacturing of each welded joint was performed by the aid of a welding robot. For the S960QL, the material was preheated to more than 90 °C but less than 150 °C, according to [24]. G4Si1 with a wire diameter of 1.2 mm was used as filler material for the welding process of S355J2+N and for S960QL the filler material Mn2NiCrMo was used with a wire diameter of 1.0 mm, according to DIN EN 757:1997-95. In both cases, M21-ArC-18 was used as inert gas with a flow rate of 15–18 L/min. The welding parameters for all cases are given in Table 3. For each material, the same welding parameters for each weld were used. For all welds the quality class B according to ISO 5817:2014-6 was reached.

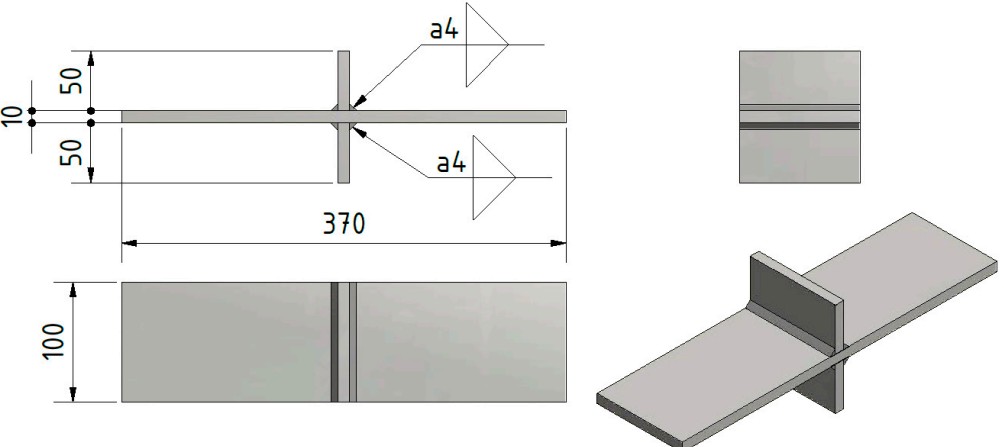

**Figure 1.** Technical drawing of welded specimen (mm).

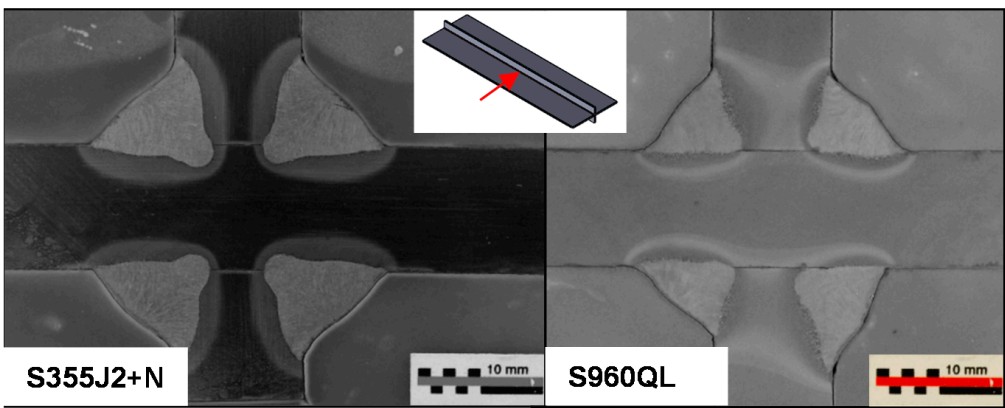

**Figure 2.** Macro graph of the weld cross sections.

**Table 3.** Process parameter for initial welding.

| Parameter | Voltage (V) | Currency (A) | Heat Input (kJ/mm) | Welding Speed (mm/s) | Efficiency (-) | Wire Feed Speed (m/min) |
|---|---|---|---|---|---|---|
| S355J2+N | 247 | 29.4 | 0.873 | 6.65 | 0.8 | 8.5 |
| S960QL | 216 | 29.4 | 1.016 | 5 | 0.8 | 9 |

Additionally, it should be mentioned that high-strength steels with yield strength of 960 MPa are susceptible to hydrogen-assisted cracking (HAC) during welding processing [25]. This should be considered in practicable applications but was not further investigated in this work.

In-situ temperature measurements were performed to determine the temperature-time-profile at the weld toe for all materials. For this, thermocouples type K with a wire diameter of 0.08 mm were used. The thermocouples were spot-welded in a distance of 0 mm, 0.5 mm, 1 mm, 2 mm, and 3 mm of the theoretical position weld toe. In total, six measurements with five thermocouples were performed in this work. Two measurements in a distance of 0.5 mm were chosen for the subsequent analyses of cooling and heating time. Figure 3 shows these temperature-time profiles.

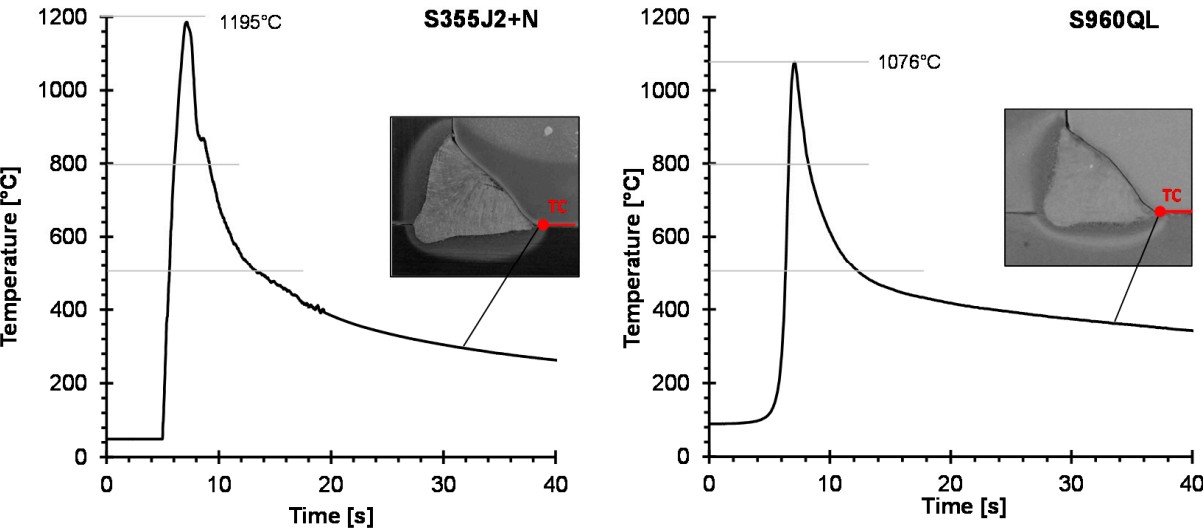

**Figure 3.** Temperature history from temperature measurement at the weld toe.

In practice, the cooling time represents an important parameter in joining of steels and enables the first verification of the welding process. To describe the cooling time, the so-called t85-time is used in fine-grained steels. This cooling time allows a qualitative statement on the microstructure of the HAZ. Besides of the performed in-situ measurement in this work, the t85-time can be calculated according to the German guidelines SEW 088 [26] and the standard DIN 1011 [27]. If the heat can dissipate from the molten pool on either side laterally in the parent material and in thickness direction the plate thickness is not negligible. In that case of 3D-dimensional heat flux, the t85-time is defined as

$$t_{85} = K_3 \, \eta \, E_1 \left( \frac{1}{500 - T_0} \right) - \left( \frac{1}{800 - T_0} \right) F_3 \tag{1}$$

with

$$K_3 = \left( 0.67 - 5 \times 10^{-4} T_0 \right) \tag{2}$$

according to SEW 088 standard, where $K_3$ is the so-called pre-heating factor, $T_0$ is the initial temperature in °C, $\eta$ is the thermal efficiency that is 0.8 for GMAW processes and $F_3$ is the heat dissipation factor that is 0.67 for fillet welds. Table 4 shows the comparison of t85-time from the measured temperature-time-profiles and the calculation according to Equation (1). The t85-time from the measurements was higher in both cases compared to the calculation.

**Table 4.** t85-time determined by temperature-measurement and analytical calculation.

| t85-Time (s) | Measurement | SEW 088 |
|---|---|---|
| **S355J2+N** | 3.79 | 2.47 |
| **S960QL** | 4.21 | 3.61 |

## 3. Repair Procedure

The basis for the repair procedure is a publication by the Federal Highway Administration of the United States, derived by a workshop of experts under the supervision of Dexter and Ocel [3]. The main goal of this publication is the repair and retrofit of fatigue cracks in steel bridges. Besides the repair welding, bolted doubler plates and stop-hole-techniques are presented as possibility to retrofit structures. The repair procedure by welding uses several steps, illustrated in Figure 4: At first, the crack detection is necessary by visual testing (VT). The NDT-personnel should look for signs of rust and cracks in the coating surface. After that, penetrant testing (PT) or magnetic testing (MT) can be used to determine the specific length and orientation of the crack. Another possibility which is not clearly stated in the repair manual [3] is the use of ultrasonic testing (UT). The advantage is, that the depth of the crack can also be determined, as long as the crack path is only in one plate member. After the detection, the removal of the crack is performed by grinding or air arc gauging. Grinding can be used in general, whereas air arc gauging should only be used at thick plates. Between the removal passes, PT or MT should be repeated in order to follow the crack path and to obtain the information of the actual crack depth. In the case of the crack depth being below one half of the plate thickness, a one-sided repair should be made. Care should be taken by a final NDT, whether the crack has been completely removed. If this is not the case, the material removal should be made to a depth of three-quarter of the plate thickness. The first side welding should then be done according to the recommended specification of the base material manufacturer. In order to avoid a high magnitude of welding residual stresses, pre-heating in the vicinity of the repair weld is recommended. In case the necessary repair depth exceeds $0.5 \times t$, the repair also has to be done from the opposite side, again by grinding or air arc gouging. After performing the counterside welding, the weld toes should be grinded smooth to reduce the amount of sharp notches.

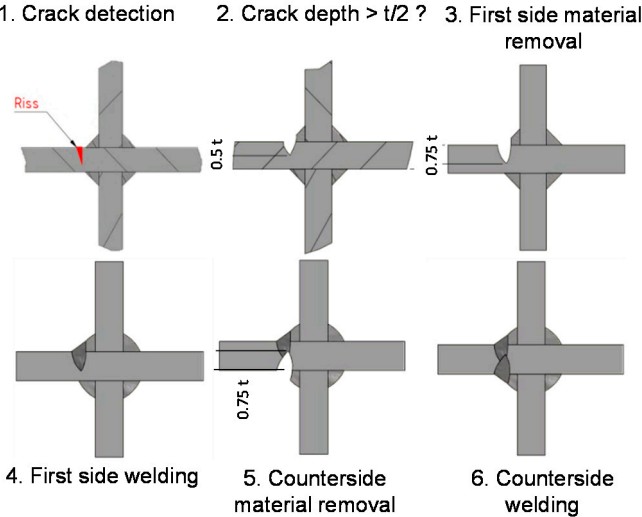

**Figure 4.** Principal procedure of the weld repair process.

Different crack depths are needed to be available for the two repair cases. For this reason, the specimens were loaded on a resonance-frequency machine RUMUL 150K (Russenberger Prüfmaschinen AG, Neuhausen am Rheinfall, Switzerland). A frequency decrease Δf was detected caused by propagating cracks associated with a stiffness decrease of the cyclic loaded specimen. A sinusoidal 4-point bending load was used in this work. Specimens with the width of 130 mm could be loaded without the crack growing through the complete specimen thickness and a comparably high weld length was available for the repair procedure. The test set-up is shown in Figure 5. Four specimens were used for a so-called beach-mark test to analyze the Δf ∼ a-correlation for different load levels. As illustrated in Figure 5, a shut-down criterion of Δf = 0.2 Hz was used for one-sided repair (a < t/2) and a shut-down criterion of Δf = 1.2 Hz was used for double-sided repairs (a > t/2). To assure that only cracks initiate at one specific weld toe, all other weld toes were treated by High Frequency Mechanical Impact (HFMI) treatment.

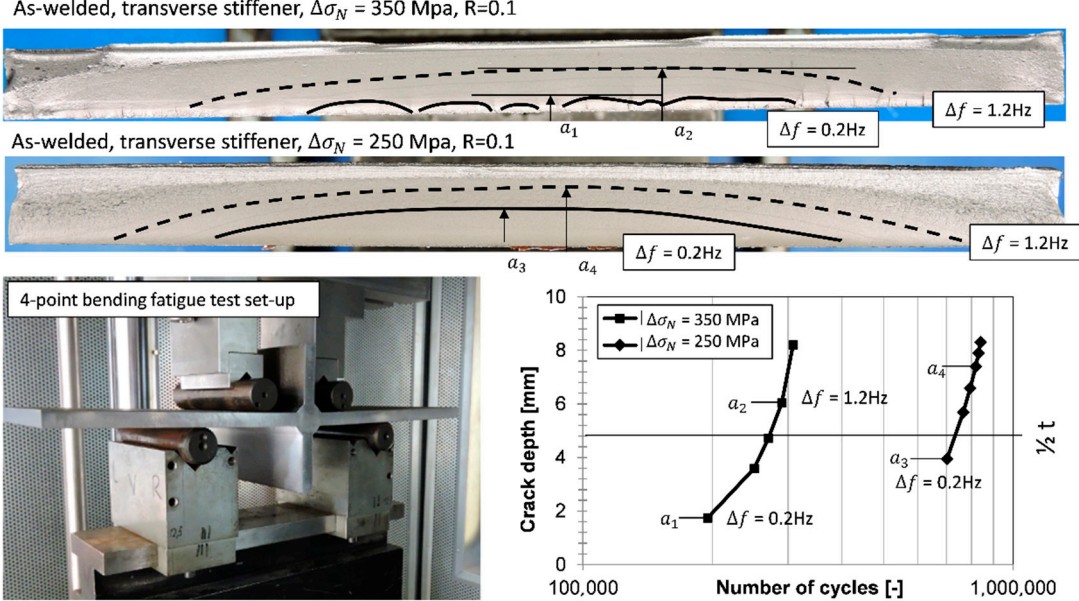

**Figure 5.** Fatigue test set-up and crack generation until a crack length of a < t/2 (one-sided repair) and a > t/2 (double-sided repair).

### 3.1. Non-Destructive Testing

The reliability of the crack detection with non-destructive testing methods is strongly dependent on the surface conditions in-situ. For this reason, the NDT-standards for MT (ISO 9934-1:2016 in conjunction with DIN EN ISO 17368:2019-05) and PT (DIN EN ISO 3452-1:2014-09 in conjunction with DIN EN ISO 23277:2015-06) give general guidelines on the necessary surface conditions for the inspection. The surface must be free from dirt, scale, rust, weld spatter, grease, oil, and other impurities and it has to be prepared in the way that relevant indications can be distinguished from false indications. In order to clarify the reason for the indication, it can be necessary to improve the surface condition by manual sanding or local grinding. For PT, precautions have to be taken not to smear surface defects by grinding. In addition, the examiner needs to consider deep grinding striations, which can lead to false indications. As the NDT-standards do not give a distinct threshold value for the surface roughness, it can be concluded, that the NDT-personnel needs to be as precise as possible in order to detect the defects. In Figure 6, an example is given on how dependent the indications are on the surface conditions.

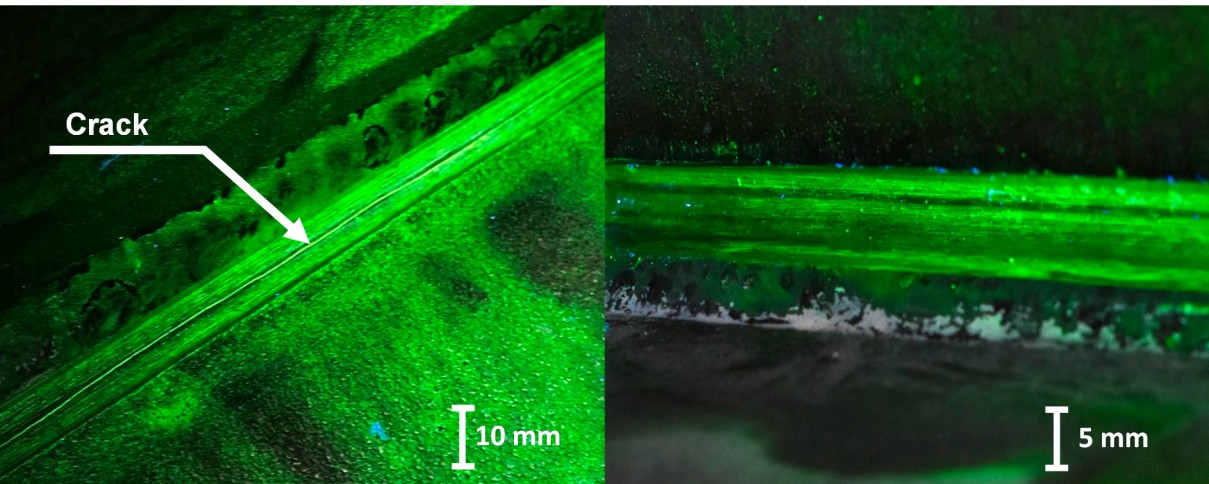

**Figure 6.** Detail of MT inspection.

On the left-hand side, a gently grinded U-groove is shown during an MT-inspection with fluorescent slurry with a clearly visible crack, which has not been fully removed. On the right-hand side, the striations introduced by strong grinding, preventing clear crack detection. A second gentle grinding pass would facilitate the interpretation.

All potentially cracked specimens were tested with PT after loading until the mentioned shut-down criterions were reached. In that way, it was assured that cracks really exists at the designated weld toe for the repair procedure and did not initiate at the other, HFMI-treated weld toes. Furthermore, these procedures proof, if such cracks are easily detectable in practical applications. However, in multiple cases, the PT lead to slightly visible or no visible cracks, illustrated in Figure 7. The percentage, were clearly or at least slightly detectable cracks were observed at the investigated specimen, are shown in Table 5. As displayed, this success rate is low for PT testing for specimen with crack depths < 0.5 t. However, PT was successfully applied for specimen with cracks depths > 0.5 t for all tested specimen. An increase of the exposure time from 5 min to the maximum of 60 min according to EN DIN ISO 3452-1:2013 leaded to no other results for PT. The specimen, where no crack was detectable with PT, MT was applied and showed for nearly all specimen clearly visible cracks. For this, is should be mentioned that all NDT methods were applied at non-loaded specimen. PT is most likely to deliver better results for specimen under mean stress or maximum stress because this leads to crack opening and to a better penetration and extraction of the PT-penetrant.

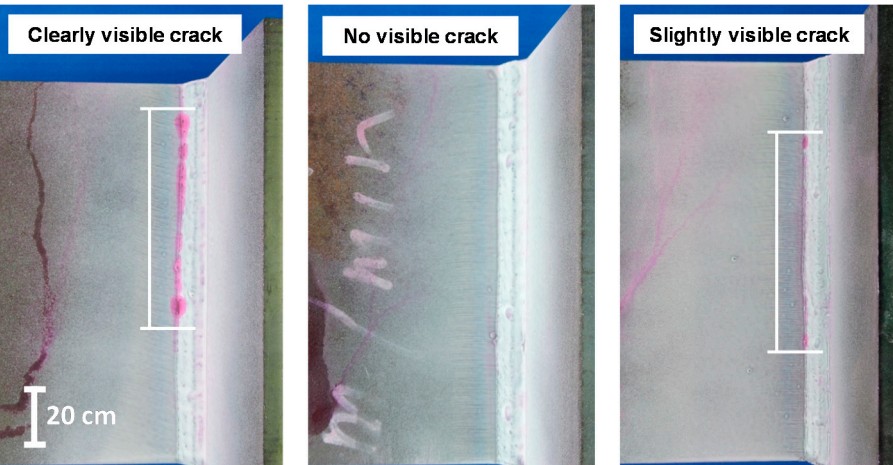

**Figure 7.** Detail of PT inspection.

**Table 5.** Percentage of detectable cracks for tested specimen.

| Success Rate (Total Number of Tested Specimen) | S355J2+N | | S960QL | |
|---|---|---|---|---|
| Crack depth | $>\frac{1}{2}$ t | $<\frac{1}{2}$ t | $>\frac{1}{2}$ t | $<\frac{1}{2}$ t |
| PT | 5.2% (19) | 100% (17) | 14.2% (21) | 100% (21) |
| MT | 83% (12) | - | 91% (12) | - |

### 3.2. Repair Welding

The repair welding was performed by gas metal arc welding (GMAW). Before welding, the material on the complete length on the specimen was removed by manual angle grinding up to a certain depth, displayed in Figure 8. Additionally, run-off-tabs were spot-welded at both sides of the specimen to avoid run-out effects and to ensure quality class B according to ISO 5817:2014-6. Crack detection by MT was performed for every specimen before the actual repair welding process, as illustrated in Figures 6 and 8. In this way, it was assured that criterion for one side repairs (crack depth < t/2) was fulfilled. Furthermore, in a depth of $0.75 \times$ t (both sided repair specimens) no cracks could be detected. The same filler material and preheat conditions are used as for the initial welding. The welding parameters are summarized in Table 6.

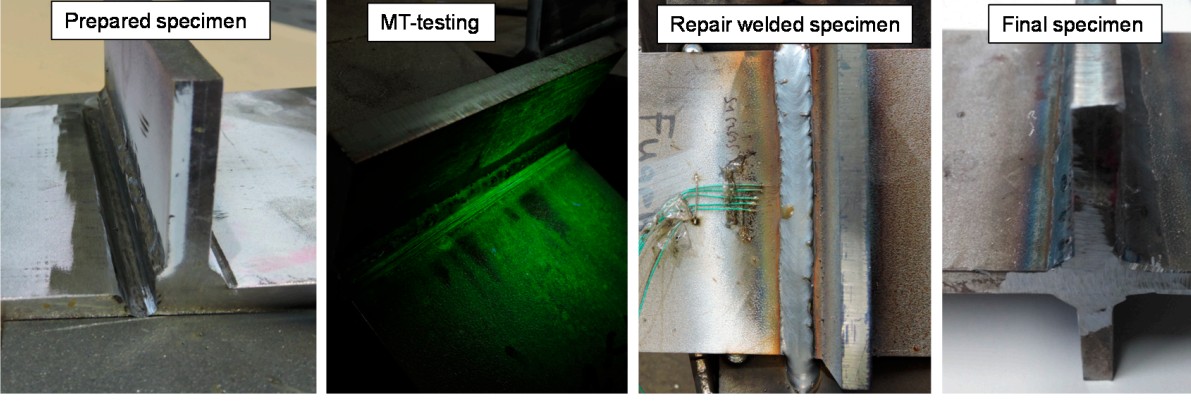

**Figure 8.** Repair procedure for investigated specimen.

**Table 6.** Process parameter for repair welding.

|  | Currency (A) | Voltage (V) | Energy per Length (J/mm) | Speed (mm/s) | Efficiency (-) | Wire Rate (m/min) |
|---|---|---|---|---|---|---|
| S355J2+N (1.pass) | 212 | 42 | 1508 | 4.72 | 0.8 | 8.5 |
| S355J2+N (2.pass) | 222 | 42 | 1579 | 4.72 | 0.8 | 8.5 |
| S960QL (1.pass) | 215 | 42 | 1529 | 4.72 | 0.8 | 8.5 |
| S960QL (2.pass) | 220 | 42 | 1565 | 4.72 | 0.8 | 8.5 |

The cross sections of the different repair cases are summarized in Figure 9. As illustrated, the penetration of the molten zone was 1 mm and 2 mm deeper than the removed material. In such cases, it may be possible that short cracks, which have not been removed by disc grinding, are molten and therefore removed. Temperature measurements were also performed during the repair welding process. The results are shown in Figure 10. As displayed the t8/5-times are higher than for the initial welding process. This can be explained by the higher heat input and the smaller welded samples which leaded to less heat dissipation during the repair welding process. The t8/5 times are compared to the guideline SWE088. The results are summarized in Table 7.

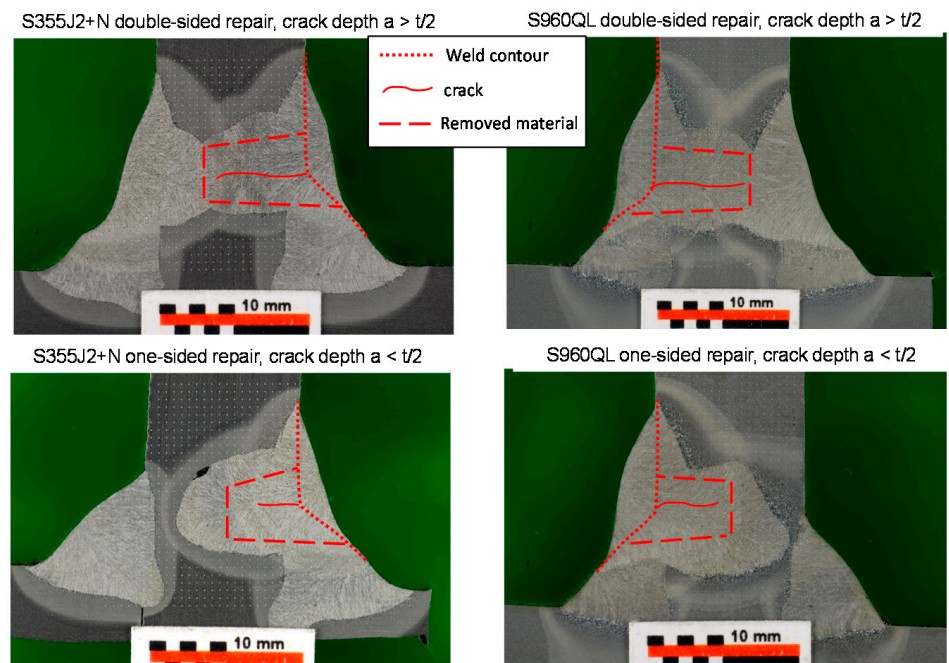

**Figure 9.** Cross section of the repaired specimen, illustrated with the original weld contour and repaired crack.

**Table 7.** t85-Time determined by temperature-measurement and analytical calculation.

| t85-Time (s) | Measurement | SEW 088 |
|---|---|---|
| S355J2+N (2.pass) | 5.66 (0.87) | 4.47 |
| S960QL (2.pass) | 6.81 (1.21) | 5.94 |

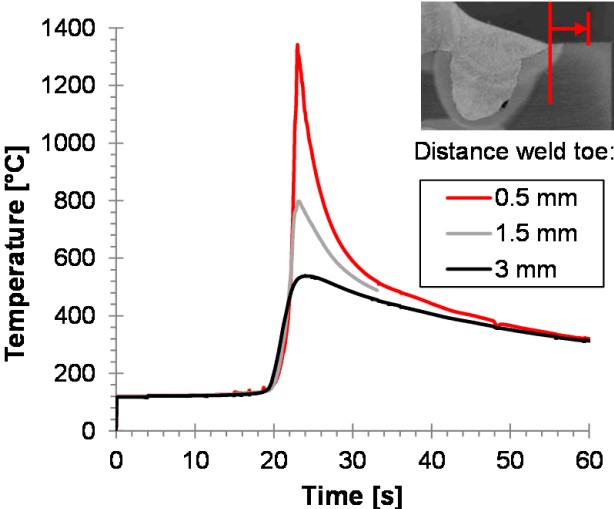

**Figure 10.** Temperature history from temperature measurement at the weld toe for the repair welding process.

## 4. Investigation of Weld Properties

### 4.1. Geometrical Properties

Recent studies showed a clear correlation between local, geometrical parameters of welded joints and their fatigue strength [28–30]. These investigations were transferred in the VOLVO quality standard that recommends specific values for weld toe radii for the first time. Besides this, recommendations on flank angles are given in standard ISO 5817:2014-06. Thus, the geometrical parameters of the weld flank angle $\theta_t$ and the weld toe radius $\rho$ according to the definition of [31] were analyzed as part of study [32]. All measurements were performed with a 3D-laser scanner and evaluated according to the so-called curvature-method [33]. Furthermore, the angle of distortion was evaluated for every specimen. Mean values and Gaussian-standard deviations are given in Table 8. As shown, similar values of $\rho$ were evaluated for every type of the investigated welded joints. However, the values of $\theta_t$ are significantly lower for the repaired (RP) conditions than for the as-welded (AW) conditions. The values of $\beta$ are similar for every investigated welded joint, but slightly higher for the RP-condition. However, compared to ISO 5817:2014-6 the angles of distortion are less than $\beta = 1°$.

**Table 8.** Mean value (μ) and standard deviation (σ) for the geometrical parameters of the investigated welds.

| System | Weld toe Radius (mm) [32] | | Flank Angle (°) [32] | | Angle of Distorsion (°) | |
|---|---|---|---|---|---|---|
| Weld | $\mu$ | $\sigma$ | $\mu$ | $\sigma$ | $\mu$ | $\sigma$ |
| S355J2+N (AW) | 0.915 | 0.356 | 39.93 | 11.28 | 0.12 | 0.051 |
| S960QL (AW) | 1.286 | 0.689 | 45.89 | 7.098 | 0.14 | 0.041 |
| S355J2+N (RP) | 0.802 | 0.209 | 18.54 | 7.170 | 0.82 | 0.125 |
| S960QL (RP) | 0.879 | 0.134 | 19.08 | 5.212 | 0.86 | 0.114 |

The assessment of the local, geometrical parameters was mainly motivated by the calculation of the stress concentration factors (SCFs) of the investigated welded joints. For the SCF calculation an approximation formulae proposed by [34] for fillet welds under tensile load and under bending load were used

$$\text{SCF}\left(\frac{t}{\rho}, \theta_t, b, s\right) = m_0 + \left(1 + m_1 \left(\frac{b}{\rho}\right)^{P_1} \left(\frac{s}{t}\right)^{P_2} + m_2 \left(\frac{t}{\rho}\right)^{P_3} + m_3 \sin(\theta_t)^{P_4}\right) \sin(\theta_t)^{P_5} \left(\frac{t}{\rho}\right)^{P_6} \quad (3)$$

where the SCF depends additionally on the throat thickness b and the root gap s, with parameters for bending load according to the Table 9.

**Table 9.** Geometrical parameters of investigated welds.

| Load | $m_0$ | $m_1$ | $m_2$ | $m_3$ | $p_1$ | $p_3$ | $p_4$ | $p_5$ | $p_6$ |
|---|---|---|---|---|---|---|---|---|---|
| tension | 1.538 | 0.621 | 1.455 | −2.933 | −1.655 | 0.208 | 1.213 | 2.086 | 0.207 |
| bending | 1.256 | 0.023 | 2.153 | −3.738 | −3.090 | 0.154 | 0.481 | 1.723 | 0.172 |

Furthermore, the SCFs were determined by direct evaluation of the ratio of the highest principle stress and nominal stress determined by 2D-Finite Element calculation according to [33]. For this, around 3000 2D-profiles were evaluated from 3D-scans and transferred into a 2D-mesh with the commercial software package HYPERMESH. Then, all profiles were automatically calculated with the ABAQUS Standard Solver. The results are summarized in Table 10. As shown, the SCFs are between 20% and 30% higher in as-welded condition than in repaired condition dependent on the evaluation method. Because of similar weld toe radii for AW and RP-condition, this difference is mainly attributed to a significantly smaller flank angle.

**Table 10.** Stress concentration factors of investigated welds [32].

| Method | Approximation [34] | | 2D-FEM [32] | |
|---|---|---|---|---|
| Weld type | $\mu$ | $\sigma$ | $\mu$ | $\sigma$ |
| S355J2+N (AW) | 2.19 | 0.218 | 1.921 | 0.170 |
| S960QL (AW) | 2.28 | 0.164 | 2.035 | 0.112 |
| S355J2+N (RP) | 1.82 | 0.161 | 1.543 | 0.173 |
| S960QL (RP) | 1.71 | 0.099 | 1.492 | 0.113 |

*4.2. Hardness*

The toughness of the heat affected zone (HAZ) is also an important factor in order to secure that neither a brittle fracture under load nor cold cracks occur [35]. For that, the maximum hardness of the HAZ should be 320 HV, 380 HV and for some special applications at a maximum of 450 HV depending on the material class. These values are also established in the standards DIN EN ISO 15614-1, API Standard 1104, and CSA Z662-07. In the case of the investigated S355J2+N steel, a maximum hardness of 320 HV10 is allowed. According to DIN EN ISO 15614-1 the maximum hardness for fine grain steels with yield strength > 890 MPa needs to be determined separately. According to [36] maximum hardness of around 433 HV could be reached for S960QL. Figure 11 displays the cross section and the hardness distribution of the weld details from each base material. All hardness measurements were performed according to DIN EN ISO 6507-1:2018-07. The hardness scale for the measurements was HV1 with an indentation point distance of 0.5 mm. The size of the indentations is less than 50 μm. The indentation distance fulfills the requirement according to ISO 6507-1:2018-07.

The HAZ of the initial weld of S355J2+N shows a hardness increase of 255 HV1 (6.2 HV1) to 275 HV1 (6.6 HV1) compared to the BM of 196 HV1 (4.1 HV1). The hardness in brackets corresponds to the standard deviation. In the case of S960QL, the hardness of the HAZ was 396 HV1 (6.9 HV1) to 420 HV1 (7.5 HV1) compared to the BM of 342 HV1 (4.1 HV1). A very similar maximum hardness with a comparable low standard deviation was measured in the HAZs in the repaired condition. These values fulfill the general requirements.

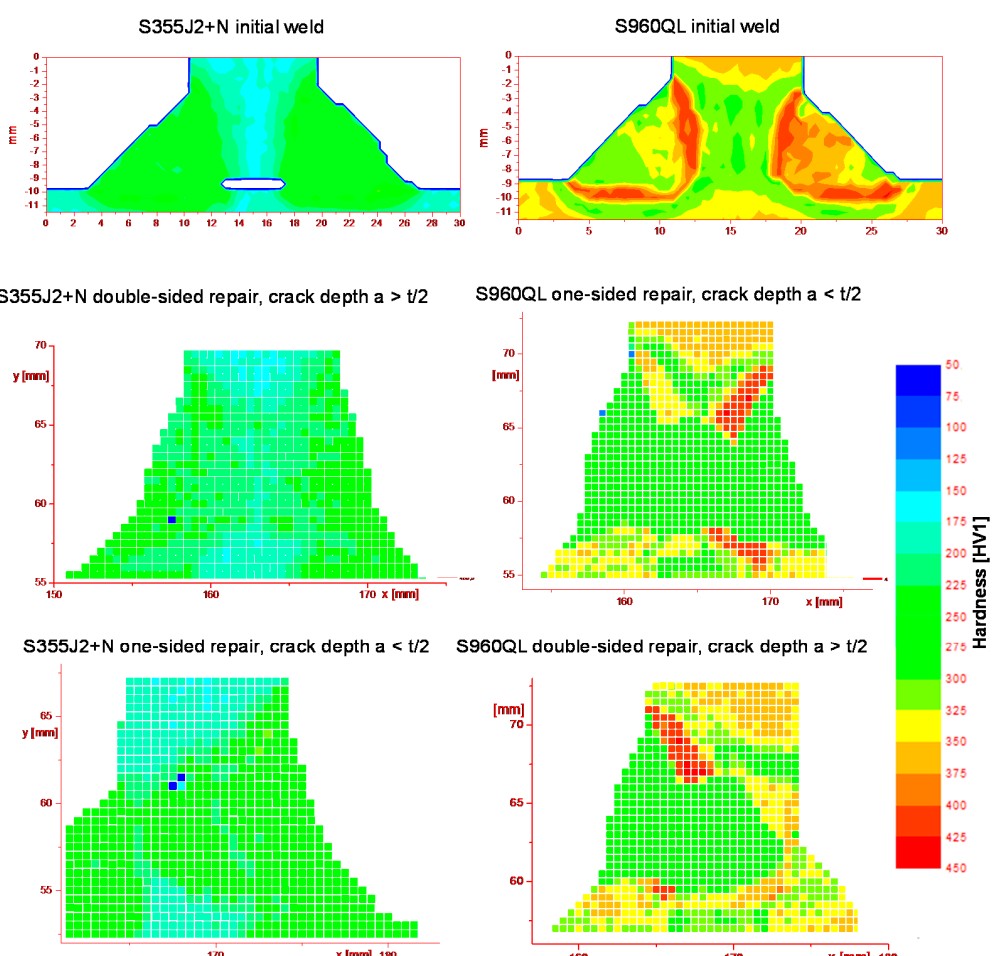

**Figure 11.** Hardness mappings for the investigated welded joints.

### 4.3. Residual Stress

It is strongly assumed that residual stresses caused by shrinking- and phase transformation effects during heating and cooling have a high influence on the fatigue behavior of welded joints. Usually, these residual stresses are handled as mean stresses in the current guidelines [23,37] or are not taken further into account for design recommendations [22]. However, low or medium residual stresses could be expected for the here-investigated, unconstrained and comparably short-width specimen according to the FKM-guideline [23] or IIW-recommendation [21]. However, this low or medium residual stress level leads to higher mean stress sensitivity and is covered by different enhancement factors. Thus, the residual stresses at the most-critical location, the weld toe was investigated experimentally for the specimen in the as-welded and the repaired condition.

The residual stresses were measured with X-ray diffraction techniques at the {211}-lattice plane with a Ca-Kr radiation. The collimator diameter for the measurement was 1 mm. For measurement preparation, the first 0.3 mm of the surface layer was removed electro-chemically to avoid measuring of overlaid compressive residual stress induced by blast-cleaning of the metal sheets before welding. The stresses in the transverse and longitudinal directions were evaluated by the $\sin(\psi)^2$-method assuming an even stress state at the surface layer.

Figure 12 illustrates the residual stress distribution from the weld toe to the base material for the as-welded (AW) and the repaired condition (RP) for both investigated materials. Furthermore, the residual stress redistribution by the separation of single specimen from the base plate was investigated. As displayed, the residual stresses in longitudinal direction are significantly higher than in transverse direction. The transverse

residual stresses close to the weld toe range from 0 MPa to 100 MPa for all investigated conditions. It is strongly assumed that negative residual stress values at a distance of >3 mm from the weld toe are related to pre-welded abrasive blast cleaning or rolling processes. Furthermore, it is shown that a higher heat input of the repair-welding process does not lead to a significantly higher residual stress level than for the original welding process.

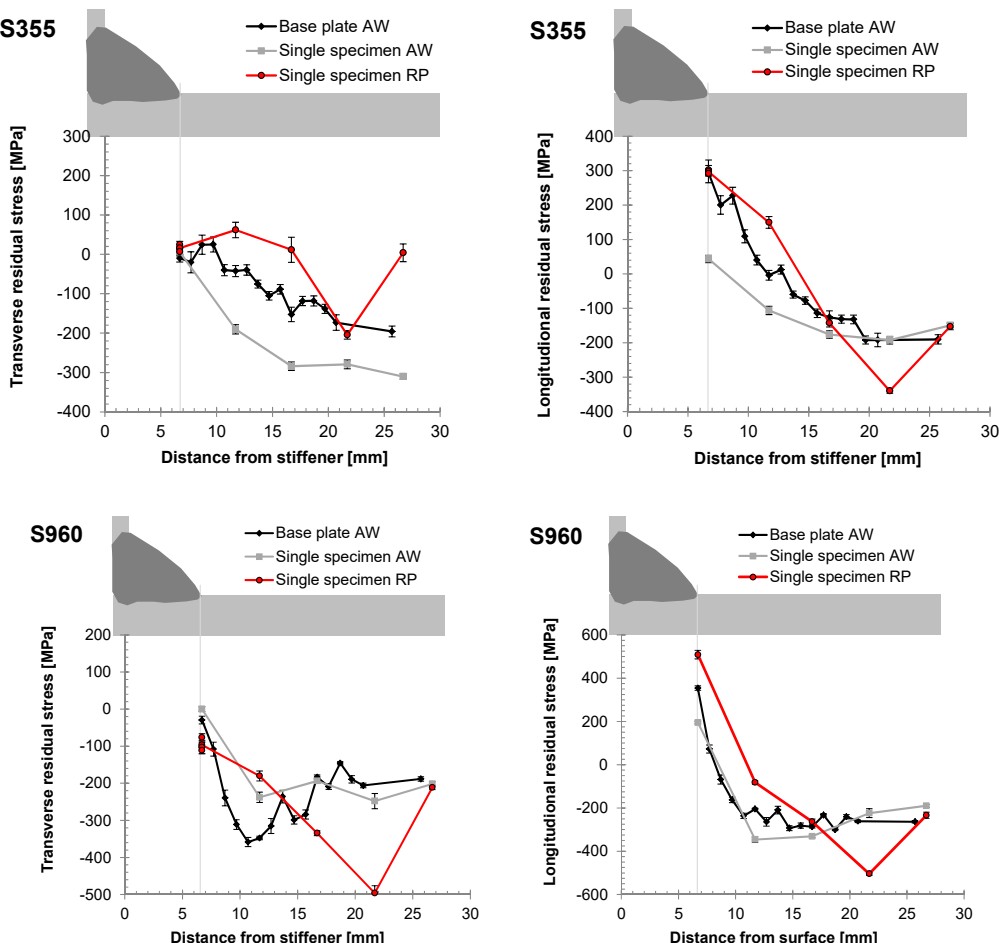

**Figure 12.** Residual stress distribution for initial welded joint and repaired welded joints.

## 5. Fatigue Analysis

The main focus of this work was the comparison of the fatigue performance of the initial-welded transverse stiffener (AW) with the repaired condition (RP). To cover both repair cases two shut down -criterions (frequency decrease) were used. $\Delta f$ = 0.2 Hz for a crack depth of a < t/2 (one-sided-repair) and $\Delta f$ = 1.2 Hz for crack depth a > t/2 (double-sided repair), as mentioned before. The corresponding fatigue results are displayed as S/N-diagramm in Figure 13 for the S355J2+N base material and in Figure 11 for the S960QL base material. No root cracks were observed at any of the investigated specimen. A clear tendency of higher fatigue life in RP-condition compared to the AW-condition is shown for both materials, but especially for the specimens made of S355J2+N. Furthermore, the fatigue test data is given in Tables 11 and 12. Specimens which did not fail from the repaired weld toe are included in this data but excluded from further evaluation. Also, run-out specimens were not included for the evaluation.

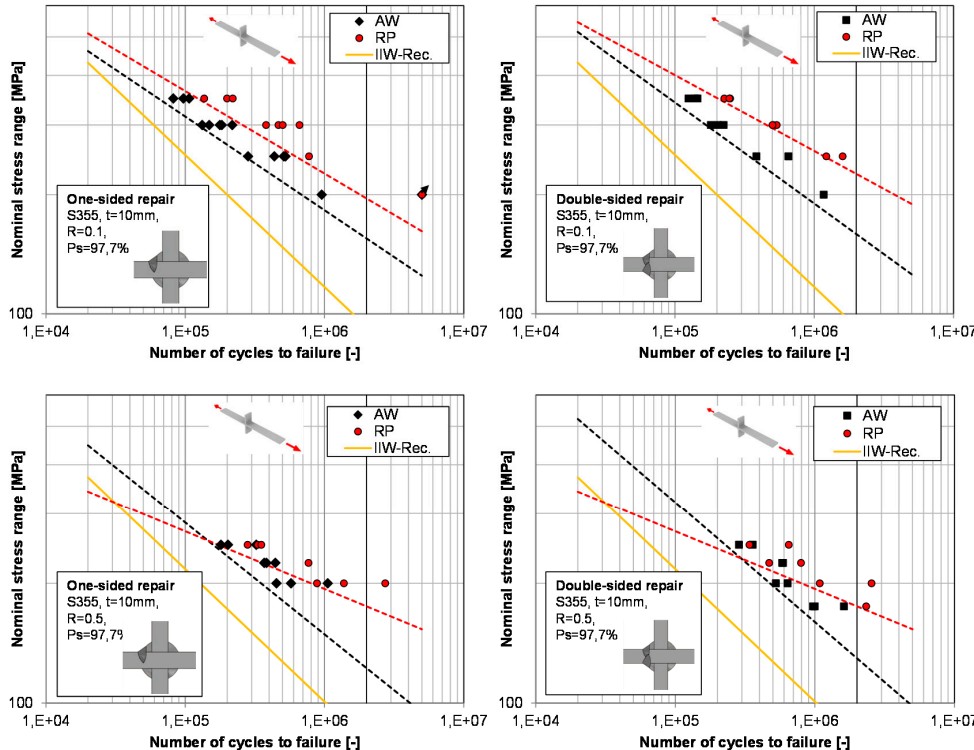

**Figure 13.** Fatigue test results for S355J2+N specimen for as-welded- (AW) and repaired (RP)-condition.

**Table 11.** Fatigue test data for S355J2+N specimen.

| | R = 0.1 | | | | | R = 0.5 | | | |
|---|---|---|---|---|---|---|---|---|---|
| ID | Δσ_N (MPa) | Δf (Hz) | N (-) AW | N (-) RP | ID | Δσ_N (MPa) | Δf (Hz) | N (-) AW | N (-) RP |
| 33 | 300 | 0.2 | 217,200 | 661,100 | 27 | 250 | 0.2 | 202,700 | 351,500 |
| 31 | 300 | 0.2 | 177,100 | 466,200 | 4 | 200 | 0.2 | 573,800 | 883,300 |
| 46 | 350 | 0.2 | 82,000 | 219,100 | 21 | 250 | 0.2 | 180,400 | 331,400 |
| 45 | 350 | 0.2 | 157,900 | 199,300 | 44 * | 250 | 0.2 | 323,000 | 280,200 |
| 43 | 200 | 0.2 | 1,248,800 | - | 37 * | 200 | 0.2 | 1,053,900 | 1,375,900 |
| 39 | 250 | 0.2 | 510,100 | - | 14 | 200 | 0.2 | 451,700 | 2,724,700 |
| 11 | 350 | 0.2 | 113,600 | 136,500 | 34 | 225 | 0.2 | 443,600 | 766,200 |
| 12 | 300 | 0.2 | 145,000 | 500,100 | 35 | 225 | 0.2 | 382,900 | - |
| 2 | 250 | 0.2 | 436,500 | 772,300 | 42 | 225 | 0.2 | 369,000 | - |
| 22 | 350 | 0.2 | 117,400 | 156,700 | 30 | 250 | 0.2 | 209,000 | 255,700 |
| 22 | 350 | 1 | 145,200 | 69,200 | 30 | 250 | 1.0 | 285,500 | 83,800 |
| 7 | 350 | 0.2 | 106,700 | 199,100 | 23 | 200 | 0.2 | 477,400 | 689,100 |
| 7 | 350 | 1 | 139,500 | 49,500 | 23 | 200 | 1.0 | 636,600 | 392,500 |
| 25 | 350 | 0.2 | 97,000 | 169,100 | 6 * | 250 | 0.2 | 271,000 | 484,200 |
| 25 | 350 | 1 | 125,100 | 76,300 | 6 * | 250 | 1.0 | 358,100 | 164,600 |
| 28 | 300 | 0.2 | 148,500 | 407,400 | 13 | 225 | 0.2 | 458,300 | 615,700 |
| 28 | 300 | 1 | 203,100 | 100,000 | 13 | 225 | 1.0 | 591,300 | 178,500 |
| 26 | 300 | 0.2 | 132,400 | 329,100 | 20 * | 225 | 0.2 | 420,400 | 292,900 |
| 26 | 300 | 1 | 181,200 | 170,900 | 20 * | 225 | 1.0 | 582,900 | 176,400 |
| 18 | 300 | 0.2 | 182,900 | 380,400 | 9 | 200 | 0.2 | 376,200 | 1,913,100 |
| 18 | 300 | 1 | 223,200 | 156,000 | 9 | 200 | 1.0 | 524,800 | 627,800 |
| 3 | 200 | 0.2 | 955,000 | 5,000,000 | 17 | 175 | 0.2 | 686,000 | - |
| 3 | 200 | 1 | 1,169,500 | - | 17 | 175 | 1.0 | 970,700 | - |
| 36 | 250 | 0.2 | 524,600 | 1,170,900 | 19 | 175 | 0.2 | 1,245,700 | 1,963,300 |
| 36 | 250 | 1 | 652,600 | 427,600 | 19 | 175 | 1.0 | 1,614,700 | 360,100 |
| 15 | 250 | 0.2 | 282,200 | 79,400 | 10 | 175 | 0.2 | 728,600 | - |
| 15 | 250 | 1 | 652,600 | 290,100 | 10 | 175 | 1.0 | 995,500 | - |

* failure from the HFMI-treated weld toe in RP condition.

**Table 12.** Fatigue test data for S960QL specimen.

| | | R = 0.1 | | | | | R = 0.5 | | |
|---|---|---|---|---|---|---|---|---|---|
| ID | $\Delta\sigma_N$ (MPa) | $\Delta f$ (Hz) | N (-) AW | N (-) RP | ID | $\Delta\sigma_N$ (MPa) | $\Delta f$ (Hz) | N (-) AW | N (-) RP |
| 85 | 500 | 0.2 | 84,100 | 37,400 | 48 * | 300 | 0.2 | 199,400 | 124,800 |
| 61 | 400 | 0.2 | 192,700 | 71,000 | 43 | 250 | 0.2 | 179,600 | 290,200 |
| 22 | 250 | 0.2 | 2,000,000 | 598,200 | 44 | 200 | 0.2 | 585,500 | 754,800 |
| 45 | 400 | 0.2 | - | 80,700 | 23 * | 300 | 0.2 | 147,400 | 104,000 |
| 4 | 500 | 0.2 | 30,500 | 45,200 | 10 | 250 | 0.2 | 223,500 | 585,600 |
| 6 | 400 | 0.2 | 214,800 | 111,300 | 11 | 200 | 0.2 | 360,800 | 615,800 |
| 17 | 250 | 0.2 | 5,000,000 | 798,600 | 30 | 300 | 0.2 | 97,200 | 391,000 |
| 12 | 400 | 0.2 | | 156,000 | 15 | 250 | 0.2 | 477,200 | 255,000 |
| 35 | 300 | 0.2 | 828,200 | 263,300 | 16 | 200 | 0.2 | 544,700 | 1,373,300 |
| 24 | 500 | 0.2 | 51,200 | 39,700 | 13 | 175 | 0.2 | 5,000,000 | - |
| 33 | 400 | 0.2 | 61,100 | 136,900 | 13 | 300 | 0.2 | 148,200 | |
| 12 | 300 | 0.2 | 980,000 | 374,800 | 19 | 300 | 0.2 | 98,900 | 353,900 |
| 8 | 500 | 0.2 | 30,900 | 50,000 | 19 | 300 | 1.0 | 138,800 | 69,800 |
| 25 | 500 | 1.0 | 44,800 | 9,700 | 34 | 250 | 0.2 | 538,600 | 355,000 |
| 25 | 400 | 0.2 | 170,800 | 114,800 | 34 | 250 | 1.0 | 715,700 | 161,100 |
| 28 | 400 | 1.0 | 201,100 | 30,000 | 20 | 200 | 0.2 | 649,100 | 2,040,800 |
| 26 | 300 | 0.2 | 817,800 | 800,000 | 20 | 200 | 1.0 | 800,800 | 428,300 |
| 26 | 300 | 1.0 | 880,800 | 82,500 | 36 | 300 | 0.2 | 233,500 | 145,400 |
| 7 | 500 | 0.2 | 26,900 | 42,700 | 36 | 300 | 1.0 | 301,100 | 42,700 |
| 18 | 500 | 1.0 | 38,300 | 12,500 | 5 * | 250 | 0.2 | 286,800 | 167,300 |
| 21 | 400 | 0.2 | 53,700 | 97,600 | 5 * | 250 | 1.0 | 413,100 | 78,400 |
| 3 | 400 | 1.0 | 78,600 | 41,000 | 29 | 200 | 0.2 | 735,000 | 5,000,000 |
| 9 | 300 | 0.2 | 273,900 | 802,200 | 29 | 200 | 1.0 | 1,077,200 | - |
| 36 | 300 | 1.0 | 368,000 | 135,300 | 40 | 300 | 0.2 | 239,700 | 394,400 |
| 31 | 500 | 0.2 | 23,500 | 91,000 | 40 | 300 | 1.0 | 289,600 | 79,700 |
| 15 | 500 | 1.0 | 34,200 | 7,500 | 37 | 250 | 0.2 | 262,800 | 1,041,100 |
| 39 | 400 | 0.2 | 126,700 | 120,900 | 37 | 250 | 1.0 | 339,400 | 193,000 |
| 39 | 400 | 1.0 | 156,700 | 23,500 | 18 | 200 | 0.2 | 673,800 | 2,051,600 |
| 38 | 300 | 0.2 | 204,000 | - | 18 | 200 | 1.0 | 858,000 | 247,800 |
| 38 | 300 | 1.0 | 226,800 | - | 28 | 300 | 0.2 | 119,600 | 90,500 |
| 32 | 500 | 0.2 | 60,000 | 92,300 | 28 | 300 | 1.0 | 162,000 | 28,400 |
| 32 | 500 | 1.0 | 75,600 | 17,000 | 42 | 250 | 0.2 | 230,700 | 5,000,000 |
| 41 | 400 | 0.2 | 243,300 | - | 42 | 250 | 1.0 | 315,800 | - |
| 41 | 400 | 1.0 | 272,000 | - | | | | | |

* failure from the HFMI-treated weld toe in RP condition.

The evaluation of the fatigue test results was performed according to DIN EN 50100-2016 to assess the FAT values of each welded joint that corresponds to the nominal stress range at $2 \times 10^6$ cycles and a survival probability ($P_s$) of 97.7% according to the IIW-recommendation. The corresponding S/N-curves are shown in Figures 13 and 14. According to Eurocode 3 [22] and IIW-Recommendation [21] a basic FAT class of 80 was proposed for the here-investigated case of a transverse stiffener without further post-weld treatment. However, in this case, an enhancement factor of f(R) = −0.4R + 1.2 according to [21] was taken into account that takes into account the comparably low residual stress level of 0.2× base material yield. For a stress ratio of R = 0.1, this leads to a FAT value of 93. The design S/N-curves evaluated with variable slope k are also displayed in Figures 13 and 14. As illustrated, in all cases, the evaluated FAT-classes lie higher than in the recommendations.

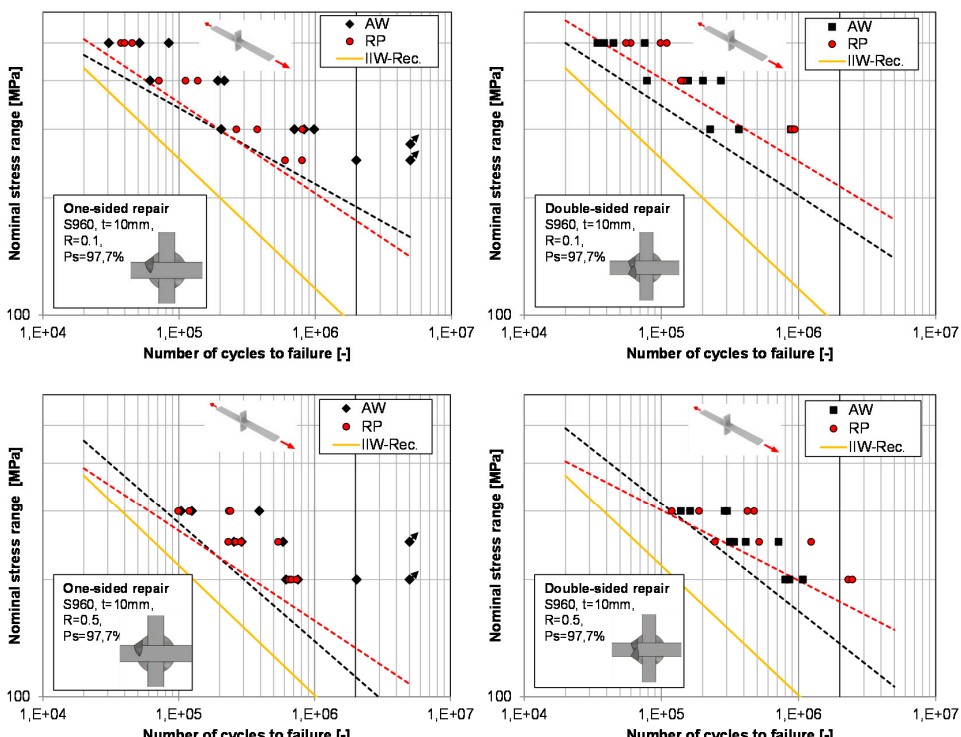

**Figure 14.** Fatigue test results for S960QL specimen for as-welded- (AW) and repaired (RP)-condition.

Furthermore, the evaluation of the FAT-classes was also performed with a fixed slope of k = 3 according to the design codes [10,11]. In this case, the FAT-classes for AW and RP conditions show lower differences of −3% to 15%. A higher difference of 25% was only observed for the base material S355J2+N and specimens loaded with R = 0.1. The comparison for both evaluations is illustrated in Figure 15.

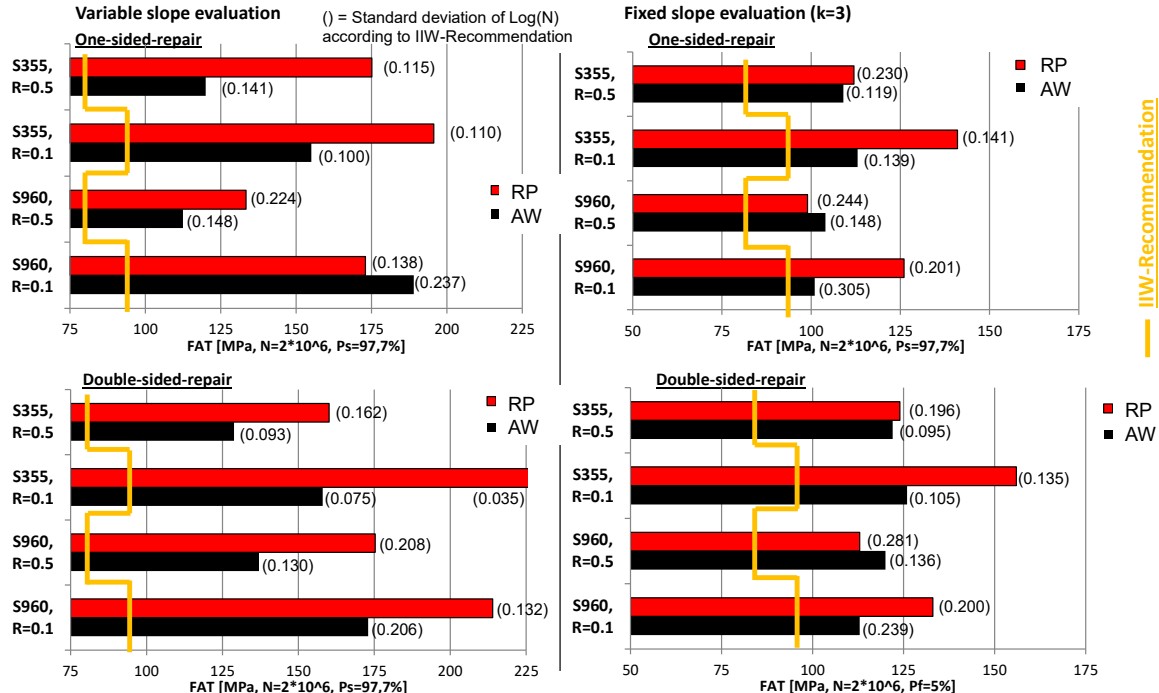

**Figure 15.** Evaluation of fatigue strength with variable and fixed slope.

Additionally, the fatigue analysis was performed according to the effective notch stress approach based on the micro support effect of Neuber [38] and interpreted on the basis of Radaj [39]. The effective notch stress was determined by Finite Element Analysis (FEA). For modelling a 2D-halfsymmetric model with a fixed weld toe radius of ρ = 1 mm was used, shown in Figure 16. Mesh size was chosen according to the recommendation of the German Welding Society [40]. The notch radius at the weld toe was meshed with three CPS8 elements (ABAQUS software package) with quadratic shape functions and a minimum size of 0.25 mm. The measured mean flank angle α, which are presented in Table 6, were used for modelling. A maximum angle of distortion of β = 1° was taken into account for the calculation of the repaired case. This represents the maximum measured distortion angle, see Table 6.

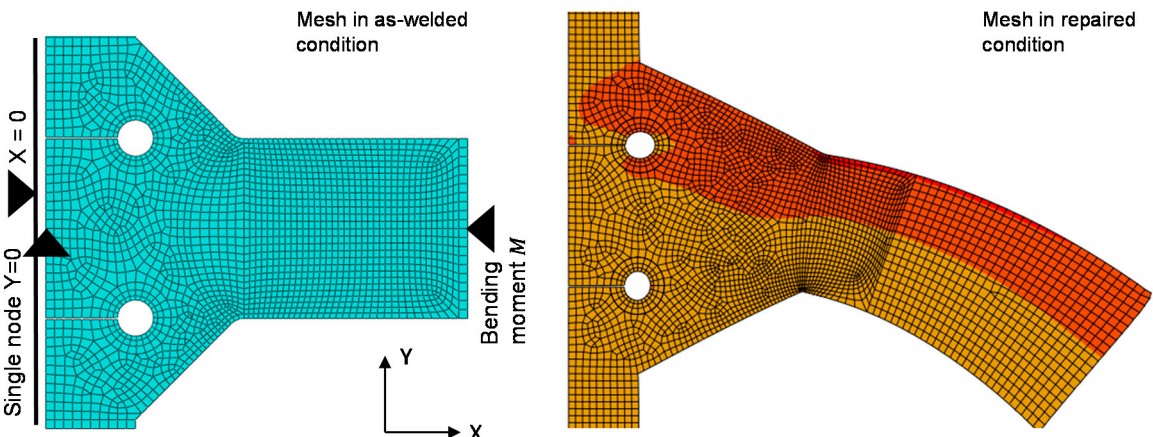

**Figure 16.** FE-mesh for the fatigue assessment according to the effective notch stress approach.

The FAT values according to the effective notch stress approach are given in Table 13. For this case, the IIW-recommendation recommends a value of FAT = 225 MPa. As shown, the determined FAT values are around 17% to 85% higher for R = 0.1 and around 5% to 51% higher for R = 0.5.

**Table 13.** FAT-classes determined according to the effective notch stress approach [39].

|  | S355J2+N | | S960QL | | IIW-Rec. [21] |
|---|---|---|---|---|---|
|  | *R* = 0.1 | *R* = 0.5 | *R* = 0.1 | *R* = 0.5 | - |
| **GZ** | 263 | 251 | 345 | 238 | |
| **RZ** | 365 | 326 | 399 | 250 | 225 |
| **RZ \*** | 380 | 340 | 417 | 262 | |

\* Taken a angle of distortion of 1° into account.

## 6. Discussion

The majority of the investigated specimen reaches at least their original life span in repaired condition. Similar results were published for similar fillet welds [13], butt joints [19] and complicated weld details like beams with gussets or weld repairs by cover plates [20] made of mild steel similar to S355 with a sheet thickness of 10 mm. A significantly higher fatigue life could be reached with a combination of weld repair and the TIG post weld treatment method [20].

The tendency of higher scatter of the fatigue tests results is shown in repaired condition, cp. standard deviation displayed in Figure 13. It is assumed that this is based on the manual repair-welding compared to the automatic original welding process. However, this trend was not observed in every test series.

The evaluated fatigue test values are significantly higher than the corresponding FAT-class of 80 MPa mentioned in the IIW recommendation [10,28]. Moreover, higher fatigue strengths in both investigated conditions are reached than in the investigation [13]. Similar transverse stiffener of S355 structural steel at R = 0.1 were tested in the AW condition and show a FAT-value of 98 MPa [41,42] and 93 MPa [15]. However, in that case, the specimens were tested under tensile load. It is assumed, that the comparably high fatigue strength in the as welded and repaired condition is based on the bending load type. Fatigue tests of identical welded joints under tensile loading confirm this assumption [43].

Similar investigations at transverse stiffeners made of S355 with the effective notch stress approach evaluate a FAT-class of 233 MPa in original condition and of 290 MPa in repaired condition [15]. It is assumed that the higher FAT-class of the here-investigated specimens is also based on the bending load type.

The residual stress level at the weld toe was low for all investigated conditions. Significant higher residual stresses are reported for GMAW welded butt joints made of S355J2 and S960QL [44]. However, no significant transverse residual stress in fillet welds was determined for SAE 1020 steel [45]. Therefore, a comparably low influence of the residual stress state on the fatigue behavior was assumed for the investigated welded joints.

A significant lower SCF was determined for the investigated welded joints in repaired condition caused by significant smaller flank angles [32]. Investigations at T-joints made of S355 show a high influence of the local weld geometry on the fatigue performance [28]. Furthermore, analytical formulae [34] for the determination of the SCF of fillet welds show a significant influence in case of constant weld toe radius. This leads to the assumption that the flank angle is the main factor for higher fatigue life of the repaired specimen in the investigated cases.

## 7. Conclusions

The aim of this work was to develop a comparably simple and reliable approach for the retrofit of fillet welds. For this, fatigue tests of transverse stiffeners made of S355J2+N and S960QL under four-point bending load under stress ratio of R = 0.1 and R = 0.5 were carried out. A crack depth of a < t/2 and of a > t/2 was controlled by a corresponding decrease of the machine test frequency. For a < t/2 an one-sided repair and for a > t/2 a repair procedure for both sides was performed. The repair contains non-destructive testing (NDT), removal of the material up to 0.75 t and a subsequent two-layer repair welding process followed by another NDT testing to assure that the specimen was crack-free. Furthermore, residual stress state, hardness, and weld geometry was investigated for the initial welded (AW) and repaired (RP) condition. The following conclusions could be made:

- No tensile residual stresses were determined at the weld toe in transverse direction for all investigated conditions.
- Hardness and microstructure are quite similar for AW and RP condition.
- The SCF is significantly lower for RP condition. This is related to a smaller flank angle of $\alpha = 40$–$45°$ compared to AW condition of $\alpha = 18$–$19°$ even if the angle of distortion is higher for RP condition.
- The majority of the repaired specimen (RP) reaches at least the fatigue life span in original condition (AW).
- All evaluated FAT values at R = 0.1 are higher for RP condition (for a fixed slope k = 3). For R = 0.5 higher FAT values in RP condition could be reached for S355 and slightly lower FAT values are reached for S960 (k = 3).
- In all cases, at least the FAT 100 according to the IIW-recommendation was reached.

According to these results, a life span for retrofitted fillet welds, repaired according to this approach, of at least the life span of the weld in original condition could be expected.

**Author Contributions:** Conceptualization, J.S.; methodology, P.L.; investigation, J.S. and P.L.; original draft preparation, J.S., P.L. and A.S.; writing—review and editing, P.K. and M.F.; project administration, P.K. and M.F.; All authors have read and agreed to the published version of the manuscript.

**Funding:** Financial support for this project was provided by IGF (German Federation of Industrial Research Associations) within the project IGF 18988 N.

**Conflicts of Interest:** The authors declare no conflict of interest and the funders had no role in the design of the study; in the collection, analyses, or interpretation of data; in the writing of the manuscript, or in the decision to publish the results.

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
