# Peer review of "Fatigue Performance of High- and Low-Strength Repaired Welded Steel Joints"

_metals, doi:10.3390/met11020293_

Round 1

Reviewer 1 Report

The submitted study is very interesting and shows well elaborated results. In order to publish the manuscript, I would like to ask you for some revisions.

  1. The previous work is very low, and isn't sufficient. It is recommended updated this section with new references.
  2. More statistical analysis should been added (CV, standard deviation, mean) of the experimental data (hardness test and fatigue test) should been added and discussed.
  3. How big was the distance of the indentation points in relation to the indent size?
  4. Regarding FEA: Which element is used? What is the type of mesh? Mesh convergences should be involved, the optimal element and size element should be specified. What is the boundary condition?
  5. The results of experimental and simulations have not been directly compared and it is difficult to evaluate them.

Author Response

Thank you for your time and for your comments. Changes in the manuscript are marked yellow. Some important information about your comments is marked green.

  1. The previous work is very low, and isn't sufficient. It is recommended updated this section with new references.

Some more references were added of the last years, see Ref [5-16]. However, it should be mentioned that for the repair welding procedure comparable less literature data is available.

  1. More statistical analysis should been added (CV, standard deviation, mean) of the experimental data (hardness test and fatigue test) should been added and discussed.

Standard deviation of the hardness and fatigue tests were added, see page 10 and Figure 12. The evaluated results were discussed in a few sentences.

  1. How big was the distance of the indentation points in relation to the indent size?

Informationen was added in section 4.2.

  1. Regarding FEA: Which element is used? What is the type of mesh? Mesh convergences should be involved, the optimal element and size element should be specified. What is the boundary condition?

Information about the Element- and mesh type were already included in the original manuscript, see page 16. The FEA was done according to the guideline in reference 27. Mesh converges are in that case not necessary because mesh size and element formulation were chosen according to [27].

Boundary conditions were visually added in Figure 13.

  1. The results of experimental and simulations have not been directly compared and it is difficult to evaluate them.

The numerical simulation according to the guideline [27] were performed to compare the FAT-classes according to the effective notch stress approach. A comparison of the recommended FAT-value according to IIW-guideline and the evaluated FAT-value was done. A direct comparison of numerical simulated stress and the real stress is possible but it needs a high effort and was not included in this work.

Reviewer 2 Report

Dear Authors,

I Have read the paper titled: "Fatigue performance of high- and low strength repaired weld steel joints".

In my opinion paper fulfills the aims and scope. However, it needs some improvement. My comments are listed below:

General remarks:

  • You have presented 29 references, in which only a few were published since 2018. It is hard to believe, that there are not relevant articles for your references. The science made big step forward last years. You should add newly published papers in your citation list.
  • I propose to change the "weld" to "welded" in the title of the paper. You have investigated all welded joints.
  • All abstract is bolded. Only word "Abstract:" should be.

Introduction:

  • This part is very weak and very short. I cannot find relevant description of scientific background for your investigations.
  • Please describe the welding technologies used for repairs and its advantages/disadvantages, e.g. preheating, temper bead welding. Than, mark advantages of your method.
  • Also, please describe the behavior of used materials during welding and problems. E.g. S355J2... grade steels could be characterized by susceptibility to cold cracking, and their repairs may require special technologies (e.g. 10.3390/app10051823). Also, in S960QL steel hydrogen-inducted cracks may occur (e.g. 10.1016/j.ijhydene.2020.05.077).
  • The novelty of your work should be strongly marked.

Materials and specimen details:

  • This paragraph requires serious improvement due to lack of many important information.
  • I propose change the name to "Materials and Specimens details". You have tested more than one specimen.
  • Table 1 - please mark the source of presented values - standard/analysis or manufacturer data? Also, the value of carbon equivalent should be marked as a weldability factor.
  • Table 2 - missing source for some columns. Have yu tested hardness?
  • How many specimens have your performed? One for each steel? It is not clear.
  • Please add information about chemical composition of wires deposited metal and their mechanical properties.
  • Line 69 "gas with a flow rate of 15-18 l/min" compared to Table 3 - it is not clear when you have used 15 l/min, and when 18 l/min.
  • Table 3 - the "Energy length" is not proper welding parameter. The proper is "heat input" with unit "kJ/mm". Please improve this issue. Also, is "speed" means "welding speed"? And "wire" means "wire feed speed"? Please use proper welding engineering nomenclature. In each specimen you have welded four welds. However, in the table, there is not any information about parematers for each tested welds. Is the same for each? Or these are avarage values?
  • Fig. 2 - These are not micro-graphs, but macro graphs. The scale bar on right picture is missing. Also, the information about etching is missing and information about preparation of specimens. In which places of the specimen, the samples were cut? How they have been performed? Have you used any standards for performing this test? Also, there are differences in the shape of welds in both specimens, which havve not been commented in the text.
  • Line 90 - change to capital letter at the beginning.

Repair procedure:

  • Line 110 - "3]Error! Reference source not found"?
  • Which version of standards were used for NDTs? Please add datas to the numbers, e.g.  ISO 9934-1:2017.
  • Fig. 7 - this is PT not MT.
  • Line 183 - please be consequent, here you used GMAW. In the rest of the text is MAG (e.g., line 63).

Rest of the text:

  • I cannot find real scientific discussion in these parts. The paragraph "4" is very short and very weak. Please compare your results with the results from literature. Please mark the advantages of your results compared to other scientiests. Please support discussion with the values.
  • Line 216 - "In the case of the here-investigated S355J2+N steel a maximum hardness of 320 HV10 is alloyed". Your materials is deffined as a material from group 1, 2 by CR ISO 15608. Following this, the Table 2 in ISO 15614-1:2017/Amd 1:2019 deffined the maxiumum hardnes for non-heat treated joints as 380 HV10. The hardness 320 HV10 is deffined for heat treated.
  • Fig. 10 - please move size of font in axis to bigger. Also, the font used in X axis and Y axis is different.

Conclusions:

  • This part is the strongest in your paper.

Author Response

Thank you for your time and for your comments. Changes in the manuscript are marked yellow. Some important information about your comments is marked green.

I Have read the paper titled: "Fatigue performance of high- and low strength repaired weld steel joints".

In my opinion paper fulfills the aims and scope. However, it needs some improvement. My comments are listed below:

General remarks:

  • You have presented 29 references, in which only a few were published since 2018. It is hard to believe, that there are not relevant articles for your references. The science made big step forward last years. You should add newly published papers in your citation list.

Some more references were added of the last years, see Ref [5-14].

  • I propose to change the "weld" to "welded" in the title of the paper. You have investigated all welded joints.

Done.

  • All abstract is bolded. Only word "Abstract:" should be.

Done.

Introduction:

  • This part is very weak and very short. I cannot find relevant description of scientific background for your investigations.

Some more references about the background of this work was added.

  • Please describe the welding technologies used for repairs and its advantages/disadvantages, e.g. preheating, temper bead welding. Than, mark advantages of your method.

Advantages and disadvantages are now described more in detail in section 1.

  • Also, please describe the behavior of used materials during welding and problems. E.g. S355J2... grade steels could be characterized by susceptibility to cold cracking, and their repairs may require special technologies (e.g. 10.3390/app10051823). Also, in S960QL steel hydrogen-inducted cracks may occur (e.g. 10.1016/j.ijhydene.2020.05.077).

Some sentences were added with the mentioned references. It is now mentioned that cold cracks and hydrogen induced cracks should be considered for practical applications but was not further investigated in this work.

  • The novelty of your work should be strongly marked.

Further background, advantages of the method were added at the instroduction, see section 1.

Materials and specimen details:

  • This paragraph requires serious improvement due to lack of many important information.
  • I propose change the name to "Materials and Specimens details". You have tested more than one specimen.

Done.

  • Table 1 - please mark the source of presented values - standard/analysis or manufacturer data? Also, the value of carbon equivalent should be marked as a weldability factor.

Done.

  • Table 2 - missing source for some columns. Have yu tested hardness?

Elongation was taken from data sheet as mentioned. All other data were tested directly.

  • How many specimens have your performed? One for each steel? It is not clear.

Information were added.

  • Please add information about chemical composition of wires deposited metal and their mechanical properties.

Information from data sheet were added in Table 1 and Table 2.

  • Line 69 "gas with a flow rate of 15-18 l/min" compared to Table 3 - it is not clear when you have used 15 l/min, and when 18 l/min.

This was a range from the specimen manufacturer. Unfortunately no more information is available.

  • Table 3 - the "Energy length" is not proper welding parameter. The proper is "heat input" with unit "kJ/mm". Please improve this issue. Also, is "speed" means "welding speed"? And "wire" means "wire feed speed"? Please use proper welding engineering nomenclature. In each specimen you have welded four welds. However, in the table, there is not any information about parematers for each tested welds. Is the same for each? Or these are avarage values?

The Table was modified according to these recommendations. A sentence was added to clarify that the same welding parameters for each weld was used. Additional information: The welding parameters are settings on the welding machines. That means, they are the exact parameters as far as possible.

  • 2 - These are not micro-graphs, but macro graphs. The scale bar on right picture is missing. Also, the information about etching is missing and information about preparation of specimens. In which places of the specimen, the samples were cut? How they have been performed? Have you used any standards for performing this test? Also, there are differences in the shape of welds in both specimens, which havve not been commented in the text.

The scale bar was added at the right picture. The scale was the same for both pictures. Some sentence were added about the specimen preparation and extraction. A sentence was added to comment the differences of the welding shape of the material S960QL compared to the material S355J2+N. Only internal standard are considered for the preparations of the cross sections.

  • Line 90 - change to capital letter at the beginning.

Done.

Repair procedure:

  • Line 110 - "3]Error! Reference source not found"?

Corrected.

  • Which version of standards were used for NDTs? Please add datas to the numbers, e.g.  ISO 9934-1:2017.

Versions were added.

  • 7 - this is PT not MT.

Corrected.

  • Line 183 - please be consequent, here you used GMAW. In the rest of the text is MAG (e.g., line 63).

Corrected.

Rest of the text:

  • I cannot find real scientific discussion in these parts. The paragraph "4" is very short and very weak. Please compare your results with the results from literature. Please mark the advantages of your results compared to other scientiests. Please support discussion with the values.

Two paragraphs were added to complement this section. A direct comparison of the evaluated fatigue strength with the nominal stress and the effective notch stress approach with several references were performed.

  • Line 216 - "In the case of the here-investigated S355J2+N steel a maximum hardness of 320 HV10 is alloyed". Your materials is deffined as a material from group 1, 2 by CR ISO 15608. Following this, the Table 2 in ISO 15614-1:2017/Amd 1:2019 deffined the maxiumum hardnes for non-heat treated joints as 380 HV10. The hardness 320 HV10 is deffined for heat treated.

This was a typing error and should means: „"In the case of the here-investigated S355J2+N steel a maximum hardness of 320 HV10 is allowed”. However, the values were corrected to 380 HV10 max. hardness. Thank you very much for this commend.

  • 10 - please move size of font in axis to bigger. Also, the font used in X axis and Y axis is different.

Size was adjusted. However, the size of the X and Y axis was originally the same.

Conclusions:

  • This part is the strongest in your paper.

Round 2

Reviewer 1 Report

The paper in the revised form can be published, because it fulfills all the conditions of a valuable scientific paper.

Reviewer 2 Report

Dear Authors,

The paper has been seriously improved. Your efforts are appritiate. The introduction presents full scientific background now. Also, the novelty is clearly marked.

The description of methodology has been improved. Your results and discussion are suitable for publishing.

I propose to accept the paper in this state.

Line 143 - "Error! Reference source not found." - please improve during proofreading.
